# Bridging the Gap: Embedding Psychosocial Oncology Research into Comprehensive Cancer Care for Children and Young People

**DOI:** 10.3390/cancers17132123

**Published:** 2025-06-24

**Authors:** Ursula Margaret Sansom-Daly, Jordana Kathleen McLoone, Joanna Elizabeth Fardell, Holly Elaine Evans, Brittany Claire McGill, Eden Grace Robertson, Christina Signorelli, Sarah Ellis, Lauren Ha, Kate Hetherington, Rachel Elizabeth Houweling, Suzanne Mary Nevin, Clarissa Evelyn Schilstra, Richard Mitchell, Michelle Haber, Richard Jules Cohn, Claire Elizabeth Wakefield

**Affiliations:** 1Behavioural Sciences Unit, Discipline of Paediatrics and Child Health, School of Clinical Medicine, Faculty of Medicine and Health, UNSW, Randwick, Sydney, NSW 2031, Australia; j.mcloone@unsw.edu.au (J.K.M.); j.fardell@unsw.edu.au (J.E.F.); holly.evans@unsw.edu.au (H.E.E.); b.mcgill@unsw.edu.au (B.C.M.); eden.robertson@unsw.edu.au (E.G.R.); c.signorelli@unsw.edu.au (C.S.); sarah.ellis@unsw.edu.au (S.E.); lauren.ha@unsw.edu.au (L.H.); k.hetherington@unsw.edu.au (K.H.); r.houweling@unsw.edu.au (R.E.H.); s.nevin@unsw.edu.au (S.M.N.); c.schilstra@unsw.edu.au (C.E.S.); richard.cohn@health.nsw.gov.au (R.J.C.); c.wakefield@unsw.edu.au (C.E.W.); 2Kids Cancer Centre, Sydney Children’s Hospital, Randwick, Sydney, NSW 2031, Australia; richard.mitchell@health.nsw.gov.au; 3Sydney Youth Cancer Service, Nelune Comprehensive Cancer Centre, Prince of Wales Hospital, Randwick, Sydney, NSW 2031, Australia; 4Western Sydney Youth Cancer Service, Crown Princess Mary Cancer Centre, Westmead Hospital, Westmead, Sydney, NSW 2145, Australia; 5Children’s Cancer Institute, Randwick, Sydney, NSW 2031, Australia; mhaber@ccia.unsw.edu.au; 6Division of Quality of Life and Pediatric Palliative Care, Department of Pediatrics, Stanford University and Stanford Medicine Children’s Health, Palo Alto, CA 94305, USA

**Keywords:** pediatric and adolescent cancer care, psychosocial support, integrated care, family-centered care, quality of life

## Abstract

Cancer care has advanced dramatically in recent decades, yet psychosocial care—support for the mental, social, and emotional health of young patients and their families—remains under-resourced and often disconnected from medical treatment. Australia is building the Minderoo Children’s Comprehensive Cancer Centre, the first of its kind in the southern hemisphere, to provide world-class cancer care for children and their families, regardless of background or location. The aim of the Centre is to fully integrate psychosocial care with cutting-edge medical research and treatment, ensuring that every child receives not only the best physical care but also support for their emotional and social needs. Research shows that when psychosocial care is included, children and families experience better quality of life and improved outcomes. However, barriers such as limited funding, fragmented research, and a lack of collaboration slow the translation of research into everyday care. This commentary highlights that to truly achieve comprehensive care, it is essential to invest in partnerships, include families in research, and reform funding so that mental health support becomes a routine part of cancer treatment. This approach will help ensure that advances in science lead to real improvements in the lives of children with cancer and their families, benefiting society as a whole.

## 1. Introduction

This commentary was written, quite literally, while staring out the window at the construction site of the Minderoo Children’s Comprehensive Cancer Centre [1] (MCCCC). Due to be completed in late 2025, the building will house Australia’s first comprehensive children’s cancer center, the first of its kind in the southern hemisphere [1,2]. The energy, hope, and excitement surrounding the new building is palpable. Inside its doors, scientists, clinicians, and researchers will sit side-by-side, buoyed by a vision of collaborative, multidisciplinary, research-driven healthcare that ensures Australia’s youngest cancer patients, and their families, receive cutting-edge treatment and care regardless of where they live, their families’ financial situations, or cultural background.

The word “comprehensive”, in this instance, is doing some heavy lifting. A National Cancer Institute-designated comprehensive cancer center requires that research is comprehensively integrated into care, at all levels, and across the entire cancer care trajectory [2]. Relevant to psychosocial oncology, key hallmarks of a comprehensive cancer center include excellence in multidisciplinary cancer treatment and patient care; the development of empirically driven clinical innovations; translational science spanning preclinical science to clinical implementation; educational programs to train cancer clinicians and scientists; education and support for patients and their caregivers; and finally, a commitment to linking patients and their caregivers with primary, supportive, and palliative care services [3].

Against this backdrop, this commentary aims to examine the evolving role of psychosocial research within pediatric oncology, highlighting both its essential contributions to holistic cancer care and the unique challenges faced in translating research into practice. We discuss the integration of psychosocial research into multidisciplinary treatment models; explore barriers to implementing evidence-based psychosocial interventions; and consider the new opportunities and persistent gaps that arise as comprehensive cancer centers strive to deliver equitable, family-centered care. In doing so, we seek to underscore the importance of embedding psychosocial research and support at every stage of the cancer care continuum, ensuring that advances in medical treatment are matched by progress in addressing the psychological and social needs of young patients and their families

## 2. Comprehensive Cancer Care—Tailored to Child and Family Needs

Several notable gaps appear when considering how we might build on international definitions to develop a local model of comprehensive cancer care for children. Firstly, to date, definitions of comprehensive cancer have mostly been age-agnostic and primarily adult-oncology-focused [3,4]. Secondly, while psychosocial oncology research can play a critical role in advancing each of these clinical aspirations, most definitions of comprehensive cancer care make little reference to the important role that psychosocial care plays in patient care [3,4]. This is a critical gap that challenges the delivery of comprehensive cancer care. However, it also presents an opportunity for groundbreaking research to advance our understanding of psycho-oncology and develop clinical care.

In addition to providing gold-standard clinical care aligned with Australia’s National Cancer Plan [5], the MCCCC will formalize collaborations between three world-leading research centers—the Kids Cancer Centre at Sydney Children’s Hospital, which provides a comprehensive service for children and adolescents with cancer and hematologic diseases and conducts the largest bone marrow transplant program in the region; the Children’s Cancer Institute, which is the largest children’s cancer research facility in the southern hemisphere and an internationally recognized leader in child cancer research; and the Behavioural Sciences Unit (BSU), Sydney, Australia’s largest pediatric psychosocial oncology research team. Together, the Children’s Cancer Institute and the Kids Cancer Centre lead ZERO Childhood Cancer (ZERO) [6,7,8,9], Australia’s national child cancer precision program, which is at the international forefront of child cancer precision medicine programs and has a strong psychosocial program of research led by the BSU [6,7,8,9,10,11,12,13,14,15,16]. The BSU was incepted as a complementary entity to the Kids Cancer Centre at the Sydney Children’s Hospital in 2001 by pediatric oncologist Professor Richard Cohn AM [Member of the Order of Australia]. Professor Cohn was driven by the principle that “cure is not enough” [17] and led critical pediatric psycho-oncology work, alongside the BSU’s inaugural Director, Professor Claire Wakefield, to understand and document the experiences, suffering, and needs of patients and their families and the effect on the wider community [18,19,20,21,22,23,24,25,26,27,28]. The BSU has since developed many theoretically driven, evidence-based interventions to address these identified needs—with meaningful lived-experience input and co-designs conducted with young people and their families [29,30,31,32,33,34]. The timely investment in constructing the MCCCC is both a celebration of the progress made in children’s cancer care and an opportunity to continue to integrate psychosocial research into the care and lives of young people and families living with cancer—acknowledging how much more needs to be done to achieve “zero childhood cancer” [35]—with zero long-term complications.

Watching the MCCCC emerge before our eyes has caused us to reflect on the progress of our field, articulate tensions often felt but not discussed, and develop a roadmap for the challenges and opportunities that lie ahead. In the spirit of progress, we are using this opportunity to ask ourselves some difficult questions as we cross the threshold of our new, comprehensive children’s cancer center. Though the building itself has a structural blueprint, and the center has a clear vision, as a field in psychosocial oncology, do we have a similarly clear roadmap or vision? What does a child-friendly, family-centered model of comprehensive cancer care look like? And what can we do to bridge the gap between innovative psychosocial research and translation to sustained, meaningful changes in the healthcare system?

## 3. Truly Integrating Supportive Care

While progress has been made in *recognizing* psychosocial care as a critical component of cancer care [36,37,38,39,40], there is much-needed work to be done to reflect change in clinical practice, including protecting resources for a sufficient and skilled workforce and support of clinician–researcher roles in order to mainstream psychosocial services, ensure accessibility for all cancer patients, and advance the field [41]. Evidence-based models of mental health and psychosocial care have consistently emerged as top priorities for young people living with cancer and their families, both in Australia [42] and overseas [43,44]. And as a field, we have been able to develop psychosocial standards of care built on an extensive empirical literature base that can now drive our clinical care—and implementation and auditing tools—to help us map the path to bridge the gaps to get there [36,45].

Yet, internationally, even in high-income countries, psychosocial care often remains under-resourced in pediatric oncology [46,47,48]. Despite efforts to screen for, and prevent, mental health problems, psycho-oncology services often remain more reactive than proactive. Healthcare systems, including in Australia, also frequently do not have the necessary resourcing, managerial support, organizational ideology or culture, or governance (bureaucracy) to enable a clinician–researcher model to flourish in the psychosocial disciplines [49,50].

Nevertheless, encouraging progress is being made. Clinical trials are increasingly including quality-of-life end-points and psychosocial sub-studies in their programs of work. For example, PRISM-IMPACT was a sub-study that aimed to understand families and health professionals’ experiences with PRISM, ZERO’s first national clinical trial [6,7,8,9]. By better understanding families’ experiences associated with the delivery of precision medicine, we can then generate resources and supportive interventions that better meet their needs—including during their precision medicine trial participation [8]. Comprehensive cancer programs that complement and enhance “usual practice” are being trialed, such as the “Engage” survivorship program. Engage provides a comprehensive assessment of survivors’ medical and psychosocial concerns and facilitates access to care to survivors otherwise “lost to follow-up” through remotely delivered assessment and personalized, risk-stratified recommendations [51]. The program highlights the interrelated nature of physical and mental health problems and the need to address both simultaneously to overcome common barriers to accessing care. Yet, progress in these trials does not always translate to success in the “real world”, despite illustrating effectiveness in the research setting.

## 4. Bridging the Gap Between Psychosocial Oncology Research and Comprehensive Cancer Care

Psychosocial oncology researchers have the potential to play a critical role in translating gold-standard research into standard-of-care practices to improve the experiences of cancer patients, survivors, and their families [52]. This imperative is reflected in research funders increasingly requiring evidence of real-world impact, both anticipated and realized, reflective of the “social return on investment” of this research [53]. Patients and families have also, rightly, come to expect research to translate into real changes in their care. Yet, there remain key systemic barriers to the translation of psychosocial oncology research into comprehensive cancer care, including the academic nature of research, the fragmented research recruitment processes, and reliance on individual clinical champions (that is, clinicians who help drive research within their organization and advocate for its integration into everyday practice). First, research is inherently academic, focused on generating knowledge rather than implementing systemic change. Funding models reflect this, supporting research activities but not operational costs for integrating findings into clinical care. Without resources for training, staffing, and sustaining psychosocial care models, even well-evidenced interventions may never be adopted in practice, and the potential benefit of research for the intended population is not fully realized. Addressing this gap necessitates prioritizing and funding translational research, as well as providing adequate support for services to enact and sustain these evidence-based changes.

Second, study-specific recruitment and ethics approvals restrict generalizability. Studies are often limited to single institutions, creating isolated, institution-specific changes rather than scalable solutions. While smaller-scale proof-of-concept and pilot studies have a critical place in the scientific pipeline, reliance on single-site studies alone can ultimately limit the broader applicability of interventions. To develop interventions that are effective across jurisdictions, it is important to build on the insights gained from these initial studies and progress them toward larger, multi-site research that can address cohort- and site-based variability and support wider implementation. Moreover, inconsistent research settings and participant pools complicate the creation of universal psychosocial care models, limiting their applicability. Patients and healthcare professionals may face uncertainty when navigating varied psychosocial care practices across different institutions, highlighting the need for cohesive, system-wide integration.

Third, while local clinical champions are essential for tailoring and driving implementation, their involvement is inherently uncertain, typically hinging upon some combination of collegial goodwill, in-kind contributions, and inputs that occur after hours due to a lack of protected time to engage in research activities. Competing clinical demands, limited funding after research grants expire, and staffing changes jeopardize the longevity of psychosocial programs that depend on clinical champions.

Each of these research gaps/challenges offers an opportunity for new approaches (see Table 1). For example, to address these challenges, we have trialed several key strategies at the BSU. One solution is fostering partnerships with not-for-profit cancer-support organizations to deliver psychosocial interventions developed through academic research at scale [54,55]. These collaborations leverage community infrastructure for national delivery, though they require researchers to become advocates and network-builders, roles for which they often lack training and resources. Another solution is embedding implementation scientists within research teams to enhance dissemination and scalability. Despite their potential, implementation scientists remain underrepresented in psycho-oncology, highlighting the need for greater focus and investment [52]. Additionally, integrating patients and families as co-leaders in research can assist with policy and advocacy efforts. Patient-inclusive advisory committees have been shown by health-services researchers to improve psychosocial service availability, fostering comprehensive care [56,57,58]. Health professionals and professional associations can also play a pivotal role in addressing these gaps by advocating for dedicated funding streams and protected time for psychosocial research translation within clinical settings. Additionally, policymakers and healthcare funders could establish incentives and policy frameworks that prioritize the system-wide integration of evidence-based psychosocial care, setting new objectives for equity, sustainability, and measurable patient outcomes.

Bridging the gap between psychosocial oncology research and clinical care necessitates structural funding reforms, cross-sector partnerships, embedded implementation science, and amplified patient voices to ensure equitable, sustainable, and holistic cancer care.

## 5. Comprehensive Cancer Care for All Young Australians

At the core of our advocacy efforts, we must focus on the integration of children, young people, and their families as co-leaders in the research we conduct. Amplifying their voices in the development, conduct, and dissemination of research studies and output is critical to ensuring that we conduct meaningful, impactful research that can generate the improved psychosocial outcomes and health system change that families want to see [59]. Families’ voices are vital to generating the health system and policy change needed to implement psychosocial interventions into practice. Increasing evidence suggests that hospitals that involve patients and families in advisory committees or roles are significantly more likely to conduct psychosocial needs screening, offer more psychosocial support programs, and offer more community support partners to families to address needs [56,57,58]. The BSU has, therefore, together with the Kids Cancer Centre, been working to develop a youth-inclusive patient and family advisory committee to ensure families’ voices will be amplified to advance research and improve care.

Despite the Australian healthcare system delivering some of the best survival outcomes for children and adolescents with cancer, equity of care in childhood cancer remains a challenge [5]. Childhood cancer does not discriminate based on social class, culture, or location; so, neither should access to cancer treatment or psychosocial oncology support. The MCCCC is a critical step toward improving access to care. Yet, undeniable inequities do exist for our most vulnerable children and families—families living in rural/remote areas of Australia [60,61], those with lower socioeconomic backgrounds [62], people living with disability [63,64], gender- and sexuality-diverse young people [65,66], Aboriginal and Torres Strait Islander families [67,68], and families with culturally and/or linguistically diverse backgrounds (CALD) [43]. Over the past decade, psychosocial researchers across Australia have begun to improve reach to rural/remote families through the development and implementation of digital interventions to address psychological distress both during [69] and beyond active cancer treatment [29,70], physical activity [33], and long-term survivorship follow-up care [71].

For lower education and health literacy levels, our research has focused on improving consent processes and patient information and support resources, with the aim of empowering patients and caregivers at some of the most challenging treatment-related junctures, including in making complex treatment decisions [72] and when approaching end-of-life [73]. Yet, much child cancer research has excluded priority populations and missed explicitly examining important social determinants of health [74]. Common to the limitation sections of many psychosocial manuscripts are statements such as, “*Our sample was limited to English-speaking participants, with few perspectives from culturally and linguistically diverse (CALD) communities.*” With Australia’s rich cultural diversity, where 3 in 10 people were born overseas [75], and Australia’s Aboriginal and Torres Strait Islander peoples and communities experience documented systemic inequities/barriers to care [76], it is difficult (perhaps impossible) to develop a study that represents all unique communities. Encouragingly, however, our field has increasingly used participatory research methods involving individuals and family members with lived experience from these priority groups. Given the intersectional nature of many social determinants of health, future research will need to better leverage these methods to examine multiple, co-occurring sources of disadvantage and health inequity to truly develop models of comprehensive cancer care for all children and young people with cancer.

## 6. A New Paradigm? Researchers as Translators

As we look out over the scaffolding and cranes next door, what steps should we take to move toward comprehensive cancer care for children and young people? The building under construction provides us with an important mandate to develop—and action—a new blueprint.

Two decades ago, Sean Phipps wrote of how, as psychosocial researchers, we were “yoked” to the increasingly fast pace of medical progress and running to keep up [77]. This remains true in 2025, where precision medicine and other novel therapeutics have continued to challenge the abilities of our colleagues to keep up across all disciplines. The promise of new technologies and therapies also carries the threat that access to these new technologies may widen existing disparities. In this era of technological advancements and the emerging integration of artificial intelligence into healthcare domains, maintaining the patient- and family-centered focus of psychosocial research and ethical considerations will remain a priority for psychosocial researchers. There is untapped potential for psychosocial researchers to serve a mediating role between the lived experience of patients and families and the health-system changes that need to occur to ensure that each young person and family’s experience is better than the last.

## 7. Conclusions

To realize the potential of comprehensive cancer care, our opportunity—and moral imperative—is to consider how we translate our field’s research achievements into real-world practice change to reduce suffering for patients, families, and the health professionals who care for them. Over the past 20 years, as a field, we have been increasingly writing about the process of *research translation*. Yet, what of the potential role of the “researcher as *translator*”? As psychosocial oncology researchers, we occupy a unique and privileged position—hearing directly from children and families about their lived experiences and the “pain points” that they face across the cancer care trajectory [7,20,30,51,62]. Subtly re-envisioning the role of the psychosocial researcher as a research translator calls for us to leverage this unique position to act as conduits to translate this knowledge among key parties—not only elevating the voices of young people and families to inform our clinical practice—but also driving the changes needed at the health system and policy levels. Success in this endeavor will enable comprehensive, humanistic, and person- and family-centered cancer care. Beyond survival, what could be more important than that?

## Figures and Tables

**Table 1 cancers-17-02123-t001:** Gaps and opportunities in the integration of psychosocial research into comprehensive models of cancer care for all children/adolescents with cancer.

Translation Gap/Challenge	Opportunity
Evidence-based models of psychosocial care are identified as research priorities for young people/families, yet in practice, psychosocial care is often under-resourced and often reactive in nature.	Psychosocial research methods and principles should be integrated alongside clinical pediatric cancer research, including using lived experience and co-design methods to ensure psychosocial concerns are addressed in how medical care and clinical trials are explained and delivered.
Scientist–practitioner models that integrate research and practice are limited in psychosocial oncology due to a lack of funding for dual clinician–researcher positions and a lack of career pathways in the health system.	Competitive research funding schemes should include dedicated and specific opportunities for clinician–researchers. Likewise, a greater number of clinician–researcher positions—including for non-medical disciplines such as psychosocial/allied health and nursing disciplines—are needed to embed research translation into the health system.
Psychosocial interventions are often tested in isolation, unintegrated with the clinical care systems in which they are hoped to be integrated.	Research should focus on testing complex interventions, including more explicitly testing referral and access pathways driven by patient and family preferences.
Individual clinical champions at individual institutions can become overburdened or may not have the capacity to consistently support psychosocial oncology research.	Partnerships with the broader health system and community, including non-profit organizations, can expand the reach of psychosocial oncology research and increase access to evidence-based interventions. Future research should examine how digital technologies and artificial intelligence can be used to enhance how families can access high-quality support and information to supplement their clinical care.
The vast majority of psychosocial oncology research is conducted in high-income, English-speaking, Westernized countries, limiting the relevance and cultural appropriateness of psychosocial models/interventions. There has been little research examining how social determinants of health intersect to promote or threaten psychosocial outcomes.	Psychosocial oncology researchers should prioritize building partnerships with community groups to increase the reach and impact of their work. Researchers (and the funding schemes that fund their work) should also incorporate strategies to address current disparities in research participation/lived experience involvement, such as approaches that support the financial remuneration of consumers/participants.
The voices of young people and their families have traditionally been left out of the research development and translation pipeline.	Ensuring that research priority-setting is part of the research pipeline is critical to ensuring studies focus on what matters most to people with the lived experience of cancer. Furthermore, upskilling and reimbursing people with lived experience to contribute their expertise is a necessary step. Partnering with people with lived experience as co-experts and co-investigators from the study idea-generating phase through to implementation/dissemination is critical to ensuring their priorities remain central to the entire research translation pipeline.

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
