# Peer review of "Bridging the Gap: Embedding Psychosocial Oncology Research into Comprehensive Cancer Care for Children and Young People"

_cancers, 2025, doi:10.3390/cancers17132123_

Round 1
Reviewer 1 Report
Comments and Suggestions for Authors
Thank you for asking me to review this Commentary submission. This piece has the potential to spark some interesting and deep reflections about the current state (and in many cases, failure) of integrating true holistic care in our approach to providing services for children with cancer. It thoroughly and critically reviews where we are now at relative to where we need to go with resect to providing this care in future, and attempts in some places to lay out a road map to get there.
I do not have many suggestions for improvement, but here are two issues to consider:
- The authors state that academic research is “focused on generating knowledge rather than implement- 138 ing systemic change. Funding models reflect this, supporting research activities but not 139 operational costs for integrating findings into clinical care. Without resources for training, 140 staffing, and sustaining psychosocial care models, even well-evidenced interventions of- 141 ten remain theoretical.” The implication that the output of academic research remains “theoretical” is not quite accurate in that it does not mean that, for example, an intervention study becomes a theory unless it is translated into use. I think that the broader point is that without translation, the power of and reason for research is not utilised in practice – thus, the very population that the research is done to benefit does not reap this benefit. Perhaps simply stating that it means that this means that the potential benefit from that research goes unrealised would be better, and then offering reflections on the possible solutions (as the authors state, amongst these would be a combination of lack of research priorities and funding on translational research, and lack of funding on services to enact that translation sustainably).
- The point made in lines 141-150 is that single-site studies limit broader applicability. There is some validity to this, but the fact remains that research must start with smaller/pilot studies in order to see if broader trials are appropriate – starting with multi-site studies is going to shut-down new potential ideas because smaller/single-site research pieces that attest to potential effectiveness, a larger trial will not be funded. So although there is a place for large, multi-site studies, revising this paragraph to acknowledge the role of these smaller studies in putting forth new ideas is needed.
- On line 213, please add "peoples and" before "communities" to: and Australia’s Aboriginal and Torres Strait Islander communities...", as it is both peoples and communities with these experiences.
Reviewer 2 Report
Comments and Suggestions for Authors
This commentary takes new perspectives on psycho-social Oncology research role in the comprehensive care cancer care for children and adolescents.
The commentary is interesting, well developed and I suggest only some minor revisions to ameliorate it.
The aims of this commentary should be explained more in detail at the beginning of the commentary: i.e. the role of psycho-social research in oncology therapy, the integration of the psycho-social research in this field, the new challenges for the psycho-social oncology treatment and so on..
I suggest underlining more suggestions to ameliorate the examined gaps: how health professionals, associations, economic policy could ameliorate this situation? Which new objectives could be taken?
What actions could be developed numbering for the first one to others? What kind of directions should take the research and how ameliorate the funding situation and the communication with the Clinical perspective?
With these new reflections the paper could be more significative for all the people who work in this field.
Reviewer 3 Report
Comments and Suggestions for Authors
The article provides a timely and compelling argument for the integral role of psychosocial oncology within comprehensive cancer care models, particularly in the context of the forthcoming Minderoo Children's Comprehensive Cancer Centre (MCCCC) in Australia.
The commentary effectively defines comprehensive cancer care, highlighting historical omissions regarding pediatric and psychosocial dimensions. It champions the principle that "cure is not enough," advocating for a holistic approach that prioritizes the well-being of children, young people, and their families. The article's strengths lie in its clear vision, the relevance of its proposed solutions, and its unwavering emphasis on patient-centered care, including the empowerment of lived experience.
The article strongly asserts that psychosocial care is a "critical component of cancer care". To reflect this in clinical practice, there is a pressing need to protect resources for a sufficient and skilled workforce and to support clinician-researcher roles, thereby mainstreaming psychosocial services. The "cure is not enough" principle, championed by Professor Cohn and embodied by the BSU, elevates psychosocial care from a mere "component" to a foundational philosophical underpinning of comprehensive cancer care. This is not simply about adding services; it is about fundamentally redefining the purpose of care to encompass well-being and quality of life alongside survival. This philosophical shift is critical because it justifies resource allocation and systemic changes required to achieve true comprehensiveness. Therefore, the article positions the MCCCC as a crucial testbed for operationalizing this principle, ensuring that the new center embodies a holistic approach to pediatric cancer care.
However, the analysis also identifies significant systemic barriers to research translation, such as under-resourced psychosocial services, limitations in scientist-practitioner models, fragmented research approaches, and an over-reliance on individual clinical champions. A critical limitation is the pervasive lack of diversity and inclusivity in psychosocial oncology research, perpetuating health disparities.
The report concludes that the commentary offers a robust framework for advancing the field, articulating persistent challenges and clear opportunities. It underscores the transformative potential of the "researcher as translator" paradigm, which calls for researchers to bridge lived experiences with health-system and policy-level changes actively. This shift is essential for achieving equitable, sustainable, and truly humanistic comprehensive cancer care for all young Australians.
Strengths of the commentary's vision and arguments :
The commentary distinguishes itself through its exceptionally clear and compelling vision for the future of comprehensive cancer care for children and young people. The authors explicitly articulate their overarching goal: "to integrate psychosocial oncology research into high-quality, person-centered comprehensive cancer care for all". This vision is not abstract; it is powerfully anchored by the imminent opening of the Minderoo Children's. Comprehensive Cancer Centre (MCCCC) in late 2025. The MCCCC serves as a tangible symbol and a "catalyst for reflection," prompting the authors to pose challenging questions about the current state of psychosocial oncology and to outline a "roadmap" for future challenges and opportunities. The foundational principle, "cure is not enough," provides a strong, values-driven direction for the entire argument, emphasizing a holistic approach that extends beyond medical treatment to address the suffering and needs of patients and their families.
The authors strategically leverage the imminent completion of the MCCCC as a rhetorical device and a tangible focal point for their arguments. This allows them to anchor abstract concepts of "comprehensive care" and "integration" to a concrete, high-profile development, making their arguments more immediate, relevant, and impactful. This approach transforms the commentary into a compelling call to action, directly tied to a significant real-world event. This strategic use of a landmark event enhances the article's persuasive power, positioning it as a timely and essential guide for the MCCCC and, by extension, potentially other developing cancer centers worldwide.
A significant strength of the commentary lies in its ability to identify critical gaps and propose relevant and actionable solutions to bridge them. The authors acknowledge that psychosocial care is often under-resourced and reactive. Their proposed solutions directly address this systemic barrier by advocating for the protection of resources for a sufficient and skilled workforce and the support of clinician-researcher roles to mainstream psychosocial services.
To overcome the limitations of single-institution studies and expand reach, the article advocates for fostering partnerships with not-for-profit cancer-support organizations, thereby leveraging existing community infrastructure for broader delivery of interventions. A key and highly relevant solution proposed is embedding implementation scientists within research teams, which directly tackles the "translation gap" by focusing on how research findings can be effectively disseminated and scaled into real-world practice.
The commentary effectively identifies several critical "Translation Gap/Challenge" points that hinder the full integration of psychosocial oncology research into comprehensive cancer care. These challenges are not isolated but represent systemic barriers within the healthcare and research ecosystem. Table 1 summarizes these gaps and the corresponding opportunities discussed in the article.
The commentary articulates significant challenges and proposes concrete opportunities to bridge the identified gaps, offering a roadmap for future progress in pediatric psychosocial oncology.
The implications for clinical practice collectively point to a redefinition of what constitutes "standard of care" in pediatric oncology. This new standard must explicitly encompass proactive, integrated psychosocial support, moving beyond a reactive model where mental health problems are addressed only after they arise. Protecting resources for a sufficient and skilled psychosocial workforce and supporting clinician-researcher roles are crucial steps to mainstreaming these services as an integral part of routine care.
Clinical practice should adopt holistic approaches that simultaneously address physical and mental health problems, recognizing their deeply interrelated nature. Enhancing patient and caregiver empowerment is vital, and it can be achieved through improving consent processes and providing comprehensive patient information and support resources, particularly during challenging treatment decisions and end-of-life discussions. Establishing patient and family advisory committees within clinical settings is also critical, as such involvement has been shown to improve psychosocial service availability. This shift demands significant investment in workforce development, training, and re-evaluating clinical workflows to ensure psychosocial care is seamlessly delivered and culturally responsive.
.
Reviewer 4 Report
Comments and Suggestions for Authors
- A very well-written and extensively sourced commentary that discusses the integration of psychosocial oncology research into comprehensive cancer care. While this particular commentary focuses on the MCCCC in Australia, the conclusions and implications of the recommendations provided are broadly applicable across cultures and medical systems.
- Would recommend citation for psychosocial standards of care reference son page 3, line 102
- Very minor edit; p: 3, line 120 there is an extra space between "Engage" and "provides"
- The commentary could benefit from clear definition/description of clinical champions, as well as highlighting the specific training and examples of work of implementation scientists
- Table 1 is a particularly helpful presentation of recommended actions to integrate research into care. As an admittedly personal formatting preference, would suggest "left-aligning" the paragraphs in each column for easier readability
- Excellent the inclusion of efforts to address inequalities in the support of families from lower socioeconomic backgrounds
- Overall, an excellent and broadly applicable commentary
